# Sustained Release of Antifungal and Antibacterial Agents from Novel Hybrid Degradable Nanofibers for the Treatment of Polymicrobial Osteomyelitis

**DOI:** 10.3390/ijms24043254

**Published:** 2023-02-07

**Authors:** Yung-Heng Hsu, Yi-Hsun Yu, Ying-Chao Chou, Chia-Jung Lu, Yu-Ting Lin, Steve Wen-Neng Ueng, Shih-Jung Liu

**Affiliations:** 1Department of Orthopedic Surgery, Bone and Joint Research Center, Chang Gung Memorial Hospital, Tao-Yuan 33305, Taiwan; 2Department of Mechanical Engineering, Chang Gung University, Tao-Yuan 33302, Taiwan

**Keywords:** degradable PLGA nanofibers, osteomyelitis, fluconazole, vancomycin, ceftazidime

## Abstract

This study aimed to develop a drug delivery system with hybrid biodegradable antifungal and antibacterial agents incorporated into poly lactic-co-glycolic acid (PLGA) nanofibers, facilitating an extended release of fluconazole, vancomycin, and ceftazidime to treat polymicrobial osteomyelitis. The nanofibers were assessed using scanning electron microscopy, tensile testing, water contact angle analysis, differential scanning calorimetry, and Fourier-transform infrared spectroscopy. The in vitro release of the antimicrobial agents was assessed using an elution method and a high-performance liquid chromatography assay. The in vivo elution pattern of nanofibrous mats was assessed using a rat femoral model. The experimental results demonstrated that the antimicrobial agent-loaded nanofibers released high levels of fluconazole, vancomycin, and ceftazidime for 30 and 56 days in vitro and in vivo, respectively. Histological assays revealed no notable tissue inflammation. Therefore, hybrid biodegradable PLGA nanofibers with a sustainable release of antifungal and antibacterial agents may be employed for the treatment of polymicrobial osteomyelitis.

## 1. Introduction

Despite advances in surgery, chronic osteomyelitis treatment remains a great challenge, often associated with a significant financial burden on healthcare systems. Osteomyelitis is an acute or chronic inflammatory process affecting the bone and its structure, secondary to the infection with pyogenic organisms, including bacteria, fungi, and mycobacteria [1,2]. Certain fungal and bacterial infections can be identified by the formation of biofilms that enhance antifungal and antibiotic drug resistance [3]. Bacterial and fungal coinfection was found in a small group of patients with osteomyelitis [4,5]. *Candida albicans* and *Staphylococcus aureus* are the nosocomial pathogens frequently responsible for severe morbidity and mortality, even with appropriate treatment. Coinfection is associated with mixed biofilms, more severe clinical manifestations, and enhanced drug resistance [6,7,8]. The mixed biofilm provides microbes with a stable environment that allows them to tolerate high antimicrobial concentrations, frequently resulting in therapeutic failure [6,7]. Antifungal and antibiotic drug concentrations a hundred-fold to a thousand-fold higher than the minimum inhibitory concentration (MIC) are generally necessary to treat coinfections [9,10].

The consensus treatment for bacterial and fungal osteomyelitis often involves both surgery and administration of antimicrobial agents [11,12]. Antibiotic-loaded polymethyl methacrylate (PMMA) (also known as bone cement) beads or spacers are well accepted for the treatment of bacterial osteomyelitis to provide a highly sustained local antibiotic concentration [13,14,15]. However, the effect of their antifungal-loading on osteomyelitis treatment is not well defined because of inconsistent drug release. Although successful eradication of fungal osteomyelitis or periprosthetic joint infections has been reported, [16] there are conflicting results [17,18]. High local concentrations of antifungal and antibiotic agents following radical debridement surgery can be a novel approach for treating complex polymicrobial infections. However, to the best of our knowledge, no study has addressed the simultaneous delivery of local, sustained, multiple antimicrobial agents for the treatment of fungal and bacterial-associated coinfections.

In this study, we developed hybrid biodegradable antifungal and antibacterial poly lactic-co-glycolic acid (PLGA) nanofibers via electrospinning that provide extended release of fluconazole, vancomycin, and ceftazidime. Fluconazole is an antifungal agent used against several fungal infections, including candidiasis, blastomycosis, coccidioidomycosis, etc. [16]. Vancomycin is an antibiotic and the treatment of choice for methicillin-resistant *S. aureus* osteomyelitis [19]. Ceftazidime is an antibiotic used to treat several bacterial infections, including joint infections, meningitis, pneumonia, sepsis, etc. [20]. Electrospinning is a versatile method to produce nanofibers or nanofiber mats from diverse polymers or polymer blends. In this process, high voltage electricity (5 to 50 kV) is applied to both a liquid solution and a collector, allowing the solution to extrude from a needle to form a jet. Once the solvent has evaporated, the jet solidifies and deposits fibers on the collector [21]. Electrospinning can be used to prepare fibers with diameters ranging from tens to hundreds of nanometers, sometimes up to a few micrometers [22]. Owing to their small diameter, large surface-to-volume ratio, and 3D networks, which mimic native extracellular matrices, nanofibers or nanofibrous mats can be used for various applications [23,24,25,26]. Distinct materials have been adopted for delivering drugs and biomolecules. PLGA is a biodegradable polymer that has received approval from pharmaceutical authorities as a therapeutic vehicle because of its extraordinary biocompatibility and biodegradability [27,28].

Our aim was to manufacture and assess electrospun antimicrobial-loaded degradable PLGA nanofibers and nanofibrous mats, and determine their in vitro and in vivo antimicrobial agent discharge pattern as a possible treatment for osteomyelitis.

## 2. Results

### 2.1. Characterization of Electrospun Nanofibers

Nanofibers incorporated with biodegradable hybrid antimicrobial agents were successfully manufactured via electrospinning. Figure 1 shows the SEM images and size distributions of the electrospun nanofibers. The measured diameter range for pure PLGA nanofibers was 1.45 ± 0.15 μm. The calculated fiber diameter range was 1.05 ± 0.11 μm and 120.0 ± 37.1 nm for fluconazole-loaded and vancomycin/ceftazidime-incorporated nanofibers, respectively. The nanofiber diameters decreased with the addition of antimicrobial agents. The increase in drug concentration reduced the PLGA percentage in the solution, which became less viscous and was more easily extended by the electric force during the spinning process. Hence, the size of the electrospun nanofibers decreased.

The wetting angles of spun mats decreased with the increasing percentage of pharmaceuticals, that is, 125.6°, 107.1°, and 117.8°, for the pristine PLGA nanofibrous mats, fluconazole-, and vancomycin/ceftazidime-embedded mats, respectively (Figure 2). The fluconazole-embedded nanofibers exhibited greater hydrophilicity than the vancomycin/ceftazidime-loaded nanofibers, mainly because fluconazole is more hydrophilic than the antibiotics.

Figure 3 illustrates the stress–strain curves of the electrospun nanofibers. The ultimate tensile strength of the spun nanofibers decreased with the incorporation of antimicrobial agents. The incorporation of pharmaceuticals decreased the percentage of polymers in the nanofibers, which reduced the resistance of the fibers to external tensile loads. Thus, the measured tensile properties were compromised.

FTIR spectroscopy was performed to confirm the presence of antimicrobial agents in the electrospun nanofibrous mats. In the drug-loaded nanofibers, the absorbance at 970 cm^−1^ was promoted by the C–H bonds of vancomycin and ceftazidime [29,30]. The new vibration at 1616 cm^−1^ was caused by the –NH bond of the added pharmaceuticals [29,30,31]. Additionally, the C=C peaks at 1500 cm^−1^ were significantly developed for pharmaceutical-loaded nanofibers with respect to the peak of the pristine PLGA nanofibrous mats. The FTIR assay results confirmed the successful incorporation of antimicrobial agents into the electrospun hybrid nanofibers (Figure 4).

The thermal behaviors of pristine PLGA, vancomycin/ceftazidime-loaded PLGA and fluconazole-loaded PLGA were assessed (Figure 5a,b). The peaks of vancomycin and ceftazidime disappeared after incorporation into the PLGA matrix (Figure 5a), while a new peak at 295 °C was noted [32,33]. Meanwhile, the endothermic peak of fluconazole at 140 °C diminished after blending with PLGA (Figure 5b) [34]. These results demonstrated the successful embedding of antimicrobial agents into the PLGA nanofibers.

Table 1 lists the average entrapment efficiency of the nanofiber membranes containing vancomycin/ceftazidime or fluconazole. While the entrapment efficiencies of ceftazidime and fluconazole were high, the efficiency of vancomycin only reached 62%. This might be due to the fact that vancomycin could not be completely dissolved in HFIP during electrospinning. Drug precipitation was noted at the outlet of the spinning needle. The drug entrapped in spun nanofibers decreased accordingly.

### 2.2. In Vitro and In Vivo Release Patterns of Antimicrobial Agents

The release characteristics of fluconazole, vancomycin, and ceftazidime from nanofibrous mats are shown in Figure 6. Triphasic liberation curves showed a high release on day 1, accompanied by a slow discharge for several days. The second set of peaks was noted at day 15 for vancomycin and ceftazidime, and day 23 for fluconazole; thereafter, the release diminished gradually. The relatively small errors in the curves suggested that the antimicrobial agents were uniformly distributed in the electrospun PLGA mats. Overall, the hybrid antimicrobial agent-embedded nanofibers provided sustained release of pharmaceuticals in vitro for more than 30 days.

Moreover, the in vivo discharge patterns indicated that the drug-incorporated nanofibrous mats discharged high levels of fluconazole, vancomycin, and ceftazidime (above the MIC) for up to 56 days in vivo (Figure 7).

### 2.3. Histological Analysis

Figure 8 shows the histological images on postoperative days 1, 7, 14 and 28. The hematoxylin and eosin-stained specimens showed notable mononuclear cell infiltrates of lymphocytes, plasma cells, and eosinophils in the muscle tissues surrounding the nanofibrous mats at day 7. Since then, the number of polymorphonuclear leukocytes diminished progressively with time up to day 28 post-operation.

## 3. Discussion

Over the past three decades, the pathogenesis of osteomyelitis has been clarified, along with the identification of factors responsible for infection. In addition to surgical treatment, many antimicrobial agents have been adopted for the treatment of osteomyelitis. Bacterial and fungal coinfection has been reported in upper respiratory tract infections and cultures from distinct medical devices, including dentures, implants, endotracheal tubes, and, most commonly, catheters [35]. Although this has rarely been reported in osteomyelitis [4,5], the treatment of orthopedic coinfections remains a great challenge [36].

The major difficulty in the treatment of osteomyelitis or other co-infections is biofilm formation. Three-dimensional bacterial and fungal biofilms protect the microbial community from antimicrobial agent damage, increasing chronic infections [6]. This dominant feature contributes to therapeutic failure. In particular, mixed biofilms generated by different species, such as *Candida albicans* and *Staphylococcus*, lead to an increase in antimicrobial agent resistance by a hundred-fold of MIC [9,10]. As morbidity and mortality increase, the successful treatment of fungal and bacterial osteomyelitis has become more challenging. Lerch et al. reported that the successful treatment of *S. aureus* and *C. albicans* coinfection requires serial and radical debridement and the use of a higher dose of fluconazole (800 mg/day or 12 mg/kg body weight) than a regular dose of 400 mg/day for an extended period. However, the role of antimicrobial agent-loaded bone cement beads was not clearly described in their study [4]. The therapeutic milestones for fungal or bacterial osteomyelitis have been reported to be radical debridement and adequate systemic/local antimicrobial agents [13,14,15,36,37,38]. Recently, many advanced antibiotic delivery methods have been developed [39,40], including our previous study presenting drug-eluting degradable PLGA beads that provide a sustained release of antifungal and antibacterial agents [12,41].

Advances in nanotechnology have led to encouraging treatments for osteomyelitis. In this study, degradable PLGA nanofibers were exploited for the sustained discharge of antimicrobial agents into the target area for infection control. Due to their elevated surface-area-to-volume ratio, nanofibers offer an advantageous vehicle for the transport of water-insoluble or poorly soluble pharmaceuticals. The three-dimensional network structure of nanofibers also mimics the architecture of the extracellular matrix of natural tissues, allowing enhanced cell functionality after the incorporation of drugs and multiple factors [42]. Our previous study examined the toxicity of electrospun nanofibers using a 3-(4,5-dimethylthiazol-2-yl)-2,5-diphenyltetrazolium bromide (MTT) assay of human fibroblast proliferation [43]. The electrospun nanofibers showed no signs of cytotoxicity. Additionally, the antimicrobial agent release from the nanofibers was determined with a disk diffusion method. After the electrospinning process, the bioactivities of released antibiotics remained high, ranging from 40% to 100% [44]. Additionally, nanofibrous mats also possess the benefits of conventional solid dosage forms, including easy processing, excellent drug stability, and simple packaging/shipping [45,46]. The biodegradable PLGA/vancomycin/ceftazidime/fluconazole nanofibers developed in this study could release high concentrations of antimicrobial agents for over 30 days in vitro, which provides advantages in terms of orthopedic infection control. This study is the first to develop biodegradable hybrid antibiotic/antifungal nanofibers using an electrospinning technique that concurrently offers the sustained discharge of elevated local concentrations of vancomycin, ceftazidime, and fluconazole.

Several factors may affect the release of pharmaceuticals from PLGA matrix systems [47], including the molecular weight and hydrophilicity of incorporated drugs, the rate of aqueous medium infiltration into the matrix, and the rate of matrix erosion. In general, the release curves can be partitioned into three stages: a primary blast, diffusion-governed discharge, and degradation-controlled release [48]. After the spinning procedure, most drugs are distributed in the nanofiber volume. Drugs on the fiber surface may lead to a primary blast release in the first few days. Afterwards, drug release is mainly governed by drug diffusion and polymer degradation. Successive drug release peaks were observed at days 14, 15, and 25 for vancomycin, ceftazidime, and fluconazole, respectively. Subsequently, a persistent steady discharge of antibiotics and antifungals was observed. Due to the different characteristics of embedded vancomycin, ceftazidime, and fluconazole, the nanofibers may present distinct antimicrobial agent-release profiles. In addition, a lower amount of vancomycin was released during both the in vitro and in vivo elution processes. Due to a relatively low solubility in HFIP, vancomycin precipitation was observed during the spinning process causing an accumulation of the precipitate at the exit of the needle and a lower conveyance to the nanofiber matrix. The drug release decreased accordingly.

The advantage of local drug delivery is high, sustained, local levels of pharmaceuticals without high systemic doses, thus, minimizing systemic toxicity [49]. Intravenous administration of antimicrobial agents leads to drug concentrations that are above MIC for susceptible microorganisms, but the levels reached are lower and the period above MIC is limited [50]. Roy et al. [51] found that both intra-articular and intravenous administration of vancomycin reached therapeutic concentrations in the synovial fluid of the knee, but intra-articular delivery resulted in peak concentrations several orders of magnitude higher, and also led to therapeutic serum levels. The half-life of intra-articular-delivered vancomycin was slightly over 3 h, with concentrations persisting above therapeutic level 24 h after injection in both the joint and serum.

Traditionally, osteomyelitis is treated with 4–6 weeks of parenteral antimicrobial agents after debridement procedure [1,13,52], and the antimicrobial concentration required to eradicate microorganisms inside biofilms can be a hundred to a thousand-fold higher than the MIC. The results of our in vivo experiments showed that the released concentrations of vancomycin, ceftazidime, and fluconazole (approximately 50–100, 1000, and 1000 μg/mL, respectively) from the nanofibers was much higher than their corresponding MICs (1.0, 2.0, and 0.5 μg/mL, respectively [12]) for more than 8 weeks (Figure 7). This provides advantages for sustained local therapeutic concentrations of antimicrobial agents in osteomyelitis treatment and prophylaxis [48,53]. Additionally, the histological assay suggested significant mononuclear cell infiltrates of lymphocytes, plasma cells, and eosinophils in muscle tissues surrounding the membrane post-implantation. Nevertheless, the number of polymorphonuclear leukocytes progressively diminished after 4 weeks (Figure 8).

Despite the proven effectiveness of the hybrid antimicrobial agent-incorporated nanofibrous mats, there were limitations in this study. First, restriction lies in the limited type of animals used. Second, a non-infected animal model was used in this study. It is unknown whether the hybrid drug-eluting PLGA nanofibers will perform differently in an infected model. Finally, the relationship between these findings and osteomyelitis in humans is unclear and requires further investigation.

## 4. Material and Methods

### 4.1. Manufacturing of Hybrid Drug-Loaded Nanofibers

PLGA was used in nanofiber fabrication (LA:GA = 75:25, with a molecular weight of 76,000–115,000 Da; Sigma-Aldrich, St. Louis, MO, USA). Vancomycin hydrochloride, ceftazidime hydrate, fluconazole, and hexafluoro-2-propanol (HFIP) (Sigma-Aldrich) were used as the antimicrobial agents and solvent, respectively.

Two-layer hybrid nanofibers were prepared using a lab-scale electrospinning apparatus. To fabricate the antifungal-loaded nanofibers, 1120 mg PLGA and 560 mg fluconazole were blended with 6 mL HFIP and subsequently spun into nanofibrous mats using a syringe/needle (internal diameter of 0.42 mm) at a temperature of 25 °C and relative humidity of 65%. The mixture transport speed was 0.5 mL/h. The voltage employed was 18,000 V, and the distance between the needle and the collector was 150 mm. This was followed by electrospinning of the PLGA/vancomycin/ceftazidime nanofibers. PLGA/vancomycin/ceftazidime (1120 mg/280 mg/280 mg) was blended with 6 mL of HFIP using the same spinning parameters as those for PLGA/fluconazole nanofibers. After electrospinning, hybrid antifungal and antibacterial nanofibers were obtained, with a thickness of approximately 0.18 mm. For comparison, pure PLGA nanofibers were also prepared by mixing PLGA (1120 mg) with HFIP (6 mL) and electrospinning into nanofibrous mats.

### 4.2. Assessment of Electrospun Nanofibers

The electrospun nanofibrous mats were evaluated using a Joel JSM-7500F scanning electron microscope (SEM; Tokyo, Japan). One hundred randomly selected nanofibers from the SEM micro-image were used to determine the distribution of the nanofiber size.

A general-purpose goniometer (First Ten Angstroms, Newark, CA, USA) (N = 3) was used to evaluate the wetting angle of the nanofibrous mats.

The tensile properties of the nanofibers were evaluated using a Lloyd tensiometer (Ametek, Berwyn, PA, USA). Nanofibrous samples with dimensions of 10 mm × 50 mm were clamped between two grips with a distance of 3 cm between the grips. The sample was stretched by the top grip at a rate of 60 mm/min for a distance of 10 cm (N = 3) before the grip returned to its starting point.

The thermal properties of pristine PLGA, vancomycin/ceftazidime-loaded PLGA, and fluconazole-loaded PLGA were identified using TA-DSC25 differential scanning calorimeter (New Castle, DE, USA). The heating rate was maintained at 10 °C/min over a 30–350 °C scan range.

The spectra of the electrospun antimicrobial agent-incorporated mats were obtained using a Thermo Fisher Nicolet iS5 Fourier-transform infrared (FTIR) spectrometer (Waltham, MA, USA). The nanofibrous samples were pressed into KBr discs and evaluated under the absorption mode. The resolution was 4 cm^−1^ with 32 scans, with a 400–4000 cm^−1^ range.

### 4.3. Entrapment Efficiency

To determine the entrapment efficiency (EE%) of added drugs, the nanofibrous mats with vancomycin/ceftazidime or fluconazole were weighed accurately in triplicate and extracted in HFIP employing a magnetic stirrer at 20 rpm. The extract solution was then centrifuged at 10,000 rpm for 10 min at ambient temperature. The drug concentrations in the eluents were characterized using a Hitachi L-2200R high-performance liquid chromatograph (HPLC; Tokyo, Japan) (N = 3). The EE % was calculated by the following equation [54]:(1)$$\mathrm{EE}\left(\%\right)=\frac{W_{m}}{W_{a}}\times 100\%$$
where EE is the entrapment efficiency, W_m_ is the amount of drug measured in the nanofibrous mat, and W_a_ is the drug added in the electrospun fibers.

### 4.4. In Vitro Pharmaceutical Discharge

The discharge pattern of antimicrobial agents from drug-incorporated nanofibrous mats was assessed using an in vitro elution scheme. Nanofibrous mats with a dimension of 10 mm × 10 mm (~5 mg) were submerged in a phosphate-buffered solution of 1 mL at 37 °C. The solution was then collected and evaluated after 24 h of isothermal incubation. The buffer was refreshed every 24 h, and the process was repeated for 30 days. The concentrations of fluconazole, vancomycin, and ceftazidime in the collected solutions were determined using the HPLC assay.

### 4.5. In Vivo Investigations

All animal-correlated processes received institutional approval (CGU109-005), and all animals were cared for under the supervision of a licensed veterinarian, in line with the ARRIVE guidelines and the regulations of the Department of Health and Welfare, Taiwan.

Male Sprague Dawley rats (7-week-old, ~250 g each) were enrolled in the tests. The rats were kept in independent cages with temperature and light control and had ad libitum access to standard mouse chow and sterilized drinking water.

The rats first received general anesthesia through isoflurane inhalation provided by a vaporizer in a polymethyl methacrylate box. Anesthesia was maintained through inhalation of isoflurane via a mask. After sedation, the right thigh of each rat was depilated, cleaned using soap, and disinfected with 70% ethanol prior to surgery. The right femoral shaft was explored using an anterolateral approach under aseptic conditions. Membranes (1 cm × 2 cm) were cut from the electrospun fluconazole/vancomycin/ceftazidime-loaded nanofibers (Figure 9a) and surgically enveloped around the right femoral shaft (Figure 9b). The wound was then sealed layer-by-layer. In vivo pharmaceutical levels were assessed by collecting tissues around the nanofibers at 1, 3, 7, 14, 28, 42 and 56 days post-operation. In vivo concentrations of vancomycin, ceftazidime, and fluconazole in the collected tissue specimens were assayed using HPLC (N = 3). In addition, a tissue biopsy was performed for histological analysis at 1, 7, 14 and 28 days post-implantation.

## 5. Conclusions

We prepared degradable antimicrobial agents incorporated in PLGA nanofibers and assessed their release behaviors. The in vitro drug discharge was assessed using HPLC, whereas a rat bone model was used for the evaluation of in vivo drug elution. The results showed that all nanofibrous mats discharged effective levels of vancomycin, ceftazidime, and fluconazole in vitro for 30 days. Animal tests also showed that the nanofibers released effective concentrations of antimicrobial agents for over 56 days after surgery. Histological assays did not reveal any notable tissue inflammation. Therefore, degradable nanofibers with sustained release of antimicrobial agents may be a potential treatment for polymicrobial osteomyelitis.

## Figures and Tables

**Figure 1 ijms-24-03254-f001:**
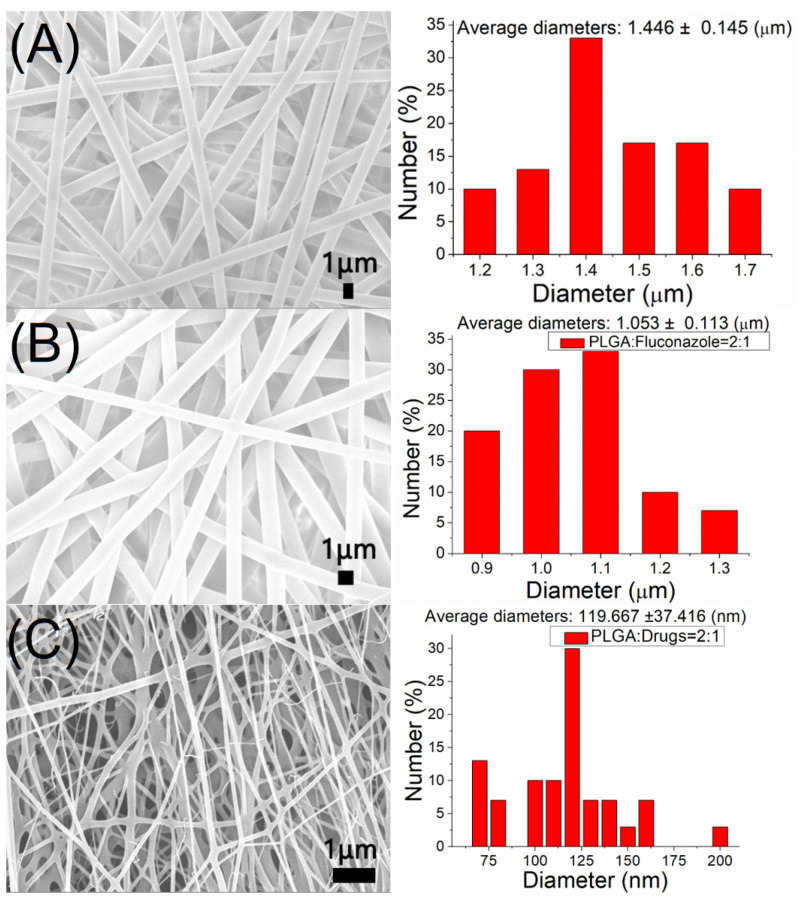
Scanning electron microscopy images and fiber diameter distributions of: (**A**) pristine poly lactic-*co*-glycolic acid (PLGA), (**B**) fluconazole-loaded nanofibers, and (**C**) vancomycin/ceftazidime-loaded nanofibers.

**Figure 2 ijms-24-03254-f002:**
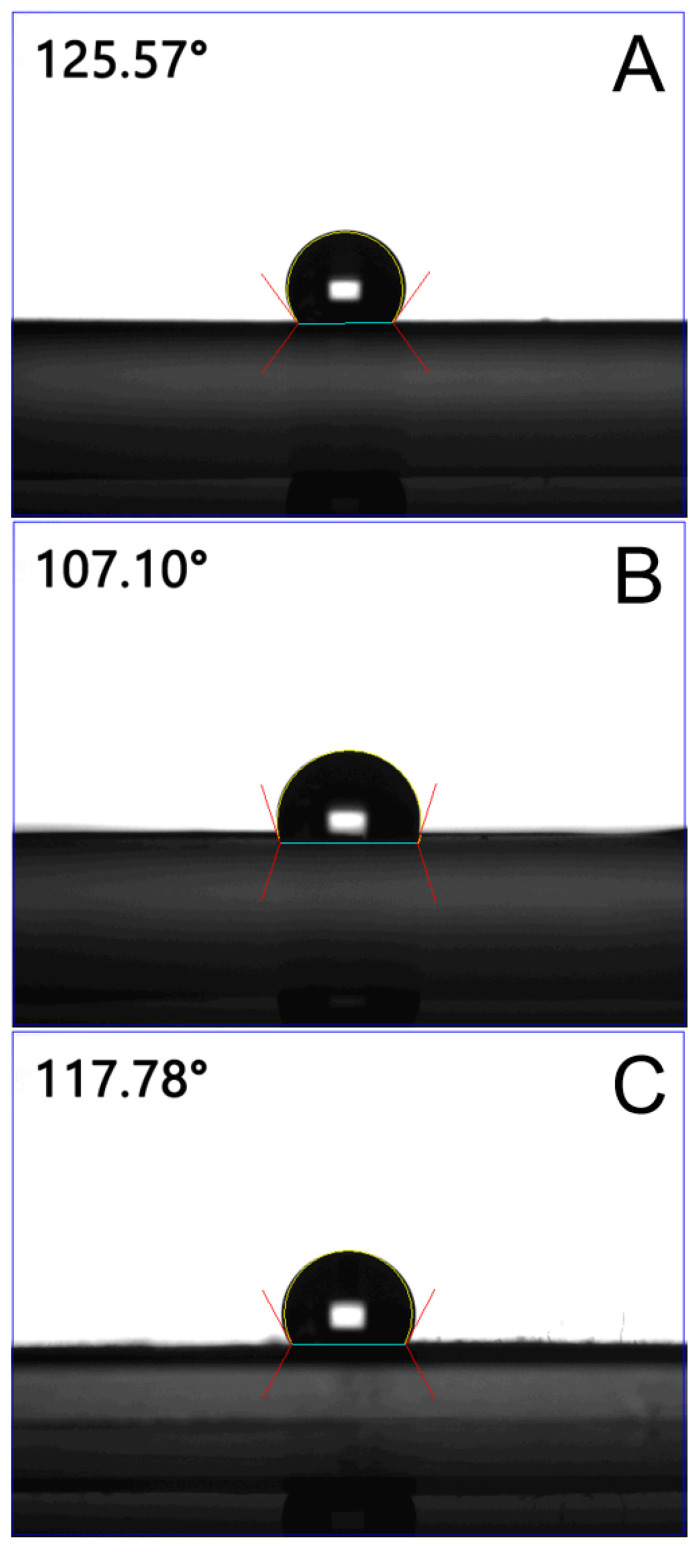
Water contact angles of: (**A**) pristine PLGA, (**B**) fluconazole-loaded nanofibers, and (**C**) vancomycin/ceftazidime-loaded nanofibers.

**Figure 3 ijms-24-03254-f003:**
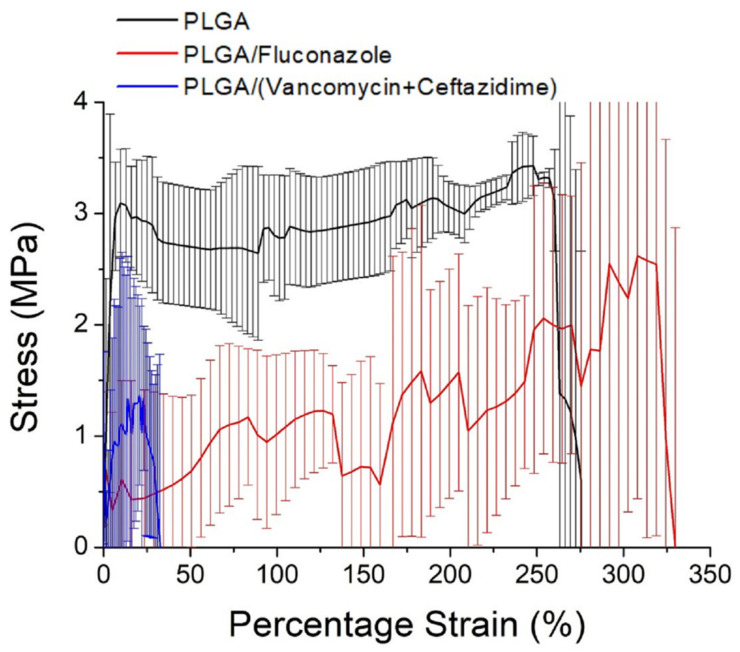
Tensile properties of electrospun drug-eluting nanofibers.

**Figure 4 ijms-24-03254-f004:**
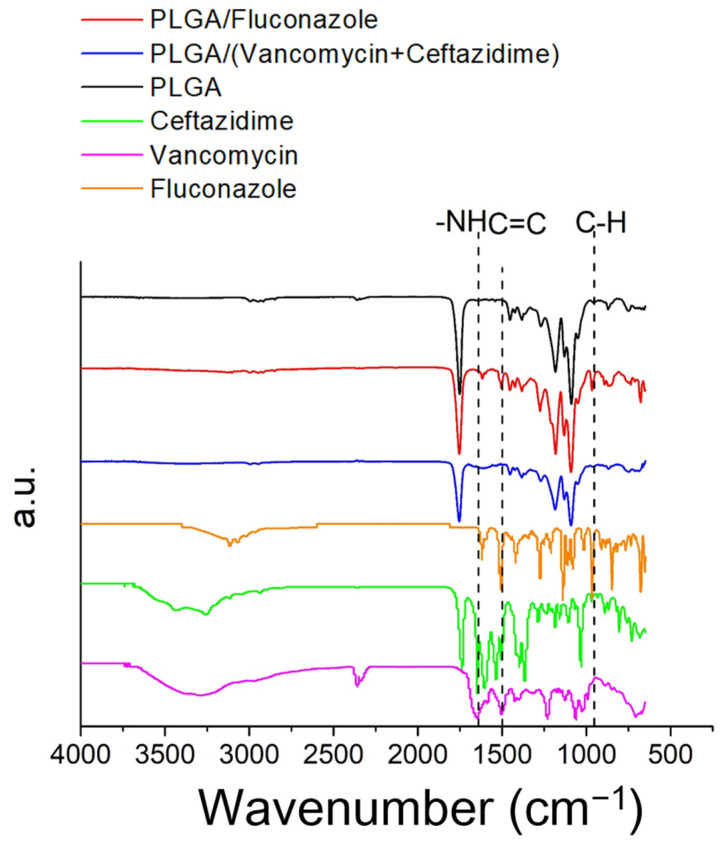
Fourier-transform infrared spectra of pure poly lactic-*co*-glycolic acid (PLGA) and antimicrobial drug-loaded PLGA nanofibers.

**Figure 5 ijms-24-03254-f005:**
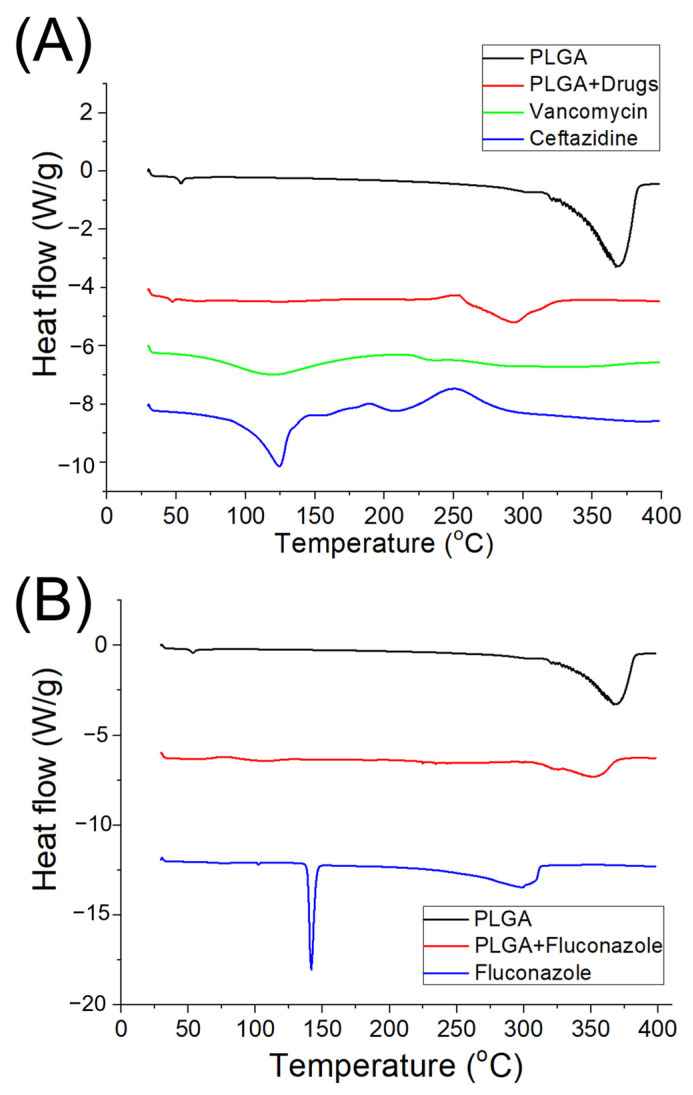
Differential scanning calorimetry assay of: (**A**) pristine poly lactic-*co*-glycolic acid (PLGA) and vancomycin/ceftazidime-loaded PLGA nanofibers, and (**B**) pure PLGA and fluconazole-loaded PLGA nanofibers.

**Figure 6 ijms-24-03254-f006:**
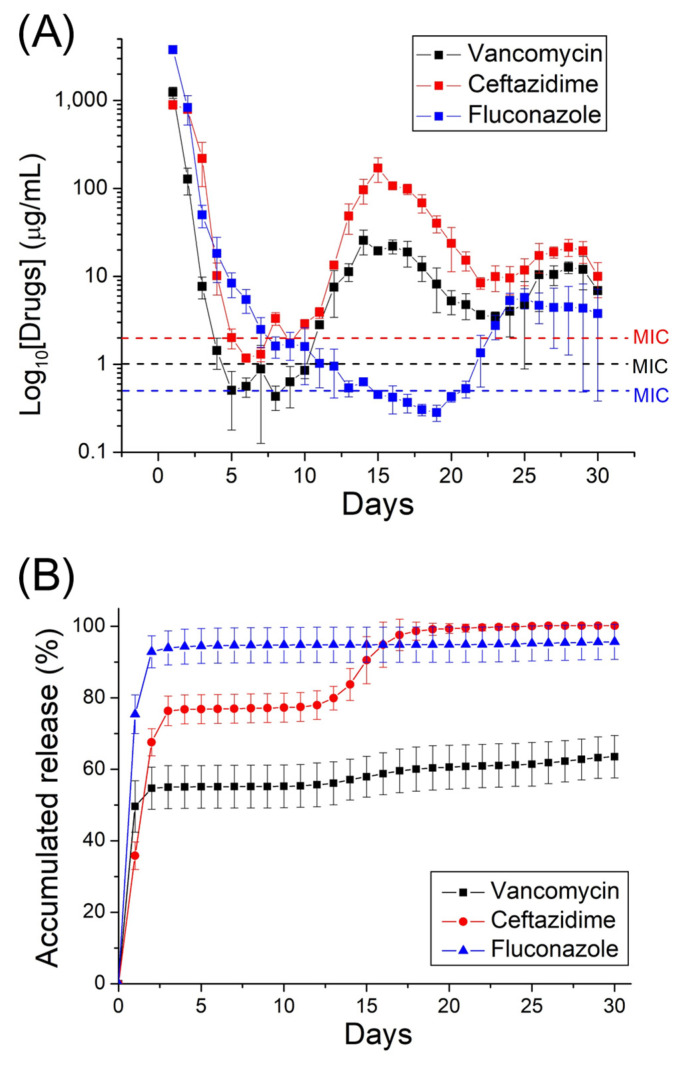
In vitro (**A**) daily, and (**B**) accumulated, release of pharmaceuticals from electrospun nanofibers (the minimum inhibitory concentrations of vancomycin, ceftazidime, and fluconazole are 1.0, 2.0, and 0.5 μg/mL, respectively).

**Figure 7 ijms-24-03254-f007:**
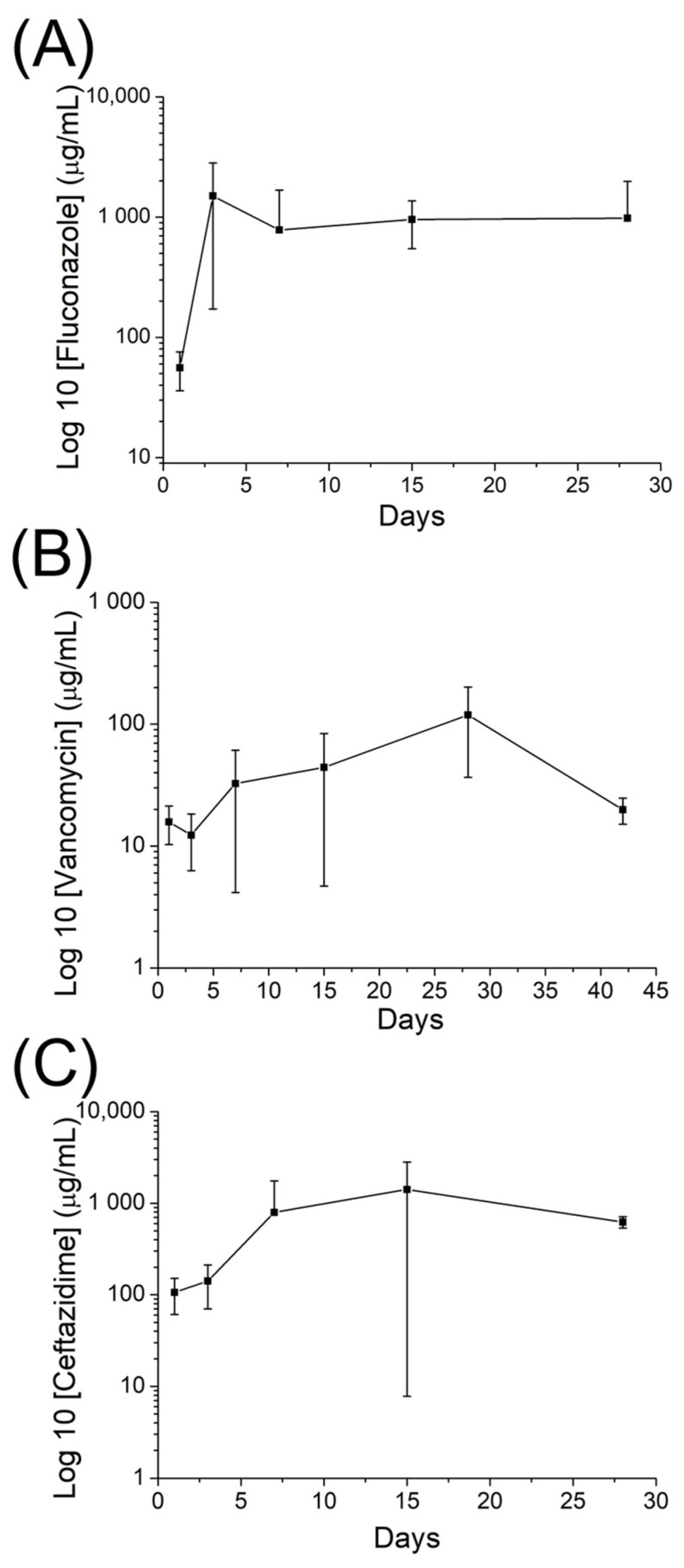
In vivo drug concentrations of (**A**) fluconazole, (**B**) vancomycin, and (**C**) ceftazidime from the nanofibers (the minimum inhibitory concentrations of fluconazole, vancomycin, and ceftazidime are 0.5, 1.0, and 2.0 μg/mL, respectively).

**Figure 8 ijms-24-03254-f008:**
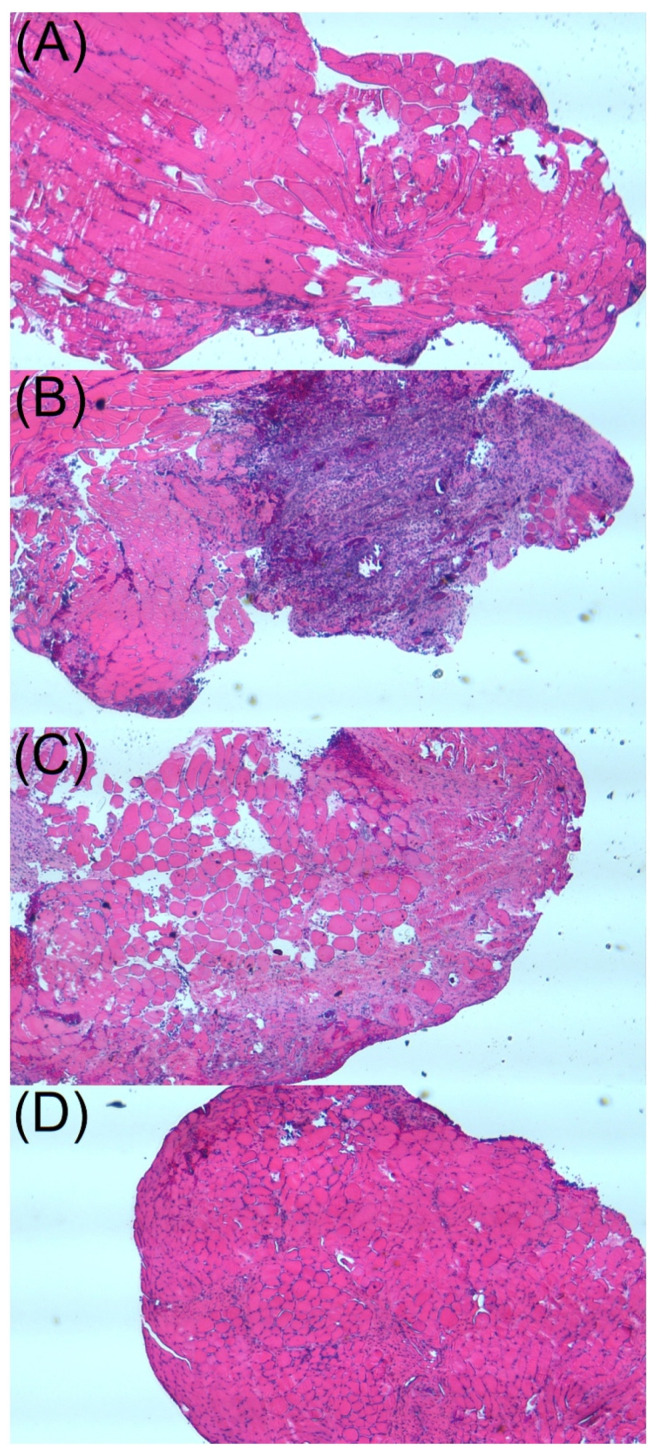
Histological analysis of muscle tissue surrounding the biodegradable membrane at (**A**) 1, (**B**) 7, (**C**) 14 and (**D**) 28 days post-implantation (200×). Microscopic examination of hematoxylin and eosin-stained specimens showed significant mononuclear cell infiltrates of lymphocytes, plasma cells, and eosinophils in the muscle tissue surrounding the membrane at day 7 after surgery. The number of polymorphonuclear leukocytes gradually decreased over time up to day 28 post-surgery.

**Figure 9 ijms-24-03254-f009:**
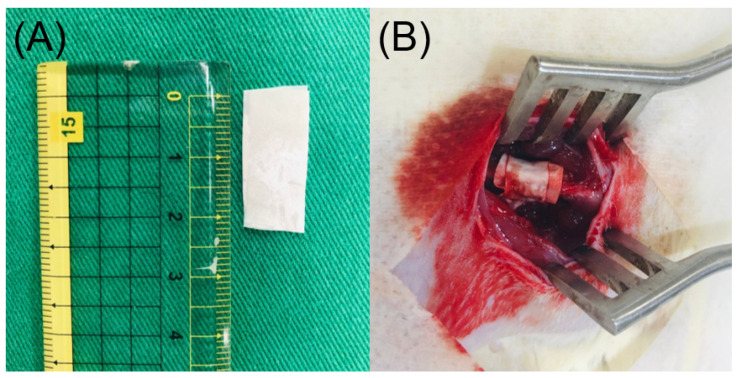
(**A**) A 1 cm × 2 cm membrane was cut from electrospun fluconazole/vancomycin/ceftazidime-loaded poly lactic-*co*-glycolic acid (PLGA) nanofibers. (**B**) The PLGA membrane was then wrapped on the right femur of the mice.

**Table 1 ijms-24-03254-t001:** Entrapment efficiencies of pharmaceuticals in the nanofibers.

Drug	Membrane Weight (mg)	Theoretical Drug Weight (μg)	Determined Drug Weight (μg)	Entrapment Efficiency (%)
Vancomycin	17.3	4325	2694.5	62.3
Ceftazidime	17.3	4325	4290.4	99.2
Fluconazole	15.8	7900	7805.2	98.8

## Data Availability

All data generated and analyzed during this study are included in the published article.

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
