# Peer review of "Sustained Release of Antifungal and Antibacterial Agents from Novel Hybrid Degradable Nanofibers for the Treatment of Polymicrobial Osteomyelitis"

_ijms, 2023, doi:10.3390/ijms24043254_

Round 1
Reviewer 1 Report
This work examines the use of PLGA nanofibers membrane incorporating fluconazole, vancomycin, and ceftazidime to treat osteomyelitis. The strategy, while not novel, has the potential to provide a more sustained and effective therapy and is still warranted. Therefore, I estimate a valuable contribution after elucidating some details:
1. Section 4.1 manufacturing of hybrid drug-loaded nanofibers- how pure PLGA nanofibers were prepared, and what are two-layer hybrid nanofibers?
2. Section 4.2 assessment of electrospun nanofibers- for tensile testing, what is the sample size, and crosshead speed? There is no indication of a number of replicate samples.
3. The entrapment efficiency should be measured.
4. Figure 3 – why the graph is not smooth, the stress is up and down, and why PLGA/Vancomycin/Ceftazidime have much lower tensile strength than PLGA/Fluconazole?
5. Figure 4 – FTIR of the pure drug (FLU, VAN, and CEF) should be analyzed.
6. The drug names in Figure 4 are abbreviations but full names are used in Figure 3.
7. Figure 5 – why does the melting peak of the drug disappear when it blends with PLGA?
Reviewer 2 Report
In this work, the authors have reported the synthesis of hybrid PLGA nanofibers for the sustained release of antifungal drugs and antibiotics. Bothe in vivo as well in vitro investigations are performed. However, prior publications some major analysis are requested as follows.
1. The authors have not calculated the drug loading capacity or percentage of release profile. without the quantitative data, it is not possible to conclude whether the release profile is sustained or not.
2. From the in vivo investigation data, it is not clear whether the authors have performed the analysis over the infected models or only local self-healing capacity of hybrid nanofibers are evaluated.
Round 2
Reviewer 1 Report
The manuscript has been sufficiently improved to warrant publication.
Reviewer 2 Report
They have addressed the concerns raised. It is now acceptable to publish.